# Sustainable and Low-Cost Electrodes for Photocatalytic Fuel Cells

**DOI:** 10.3390/nano14070636

**Published:** 2024-04-06

**Authors:** Naveed ul Hassan Alvi, Mats Sandberg

**Affiliations:** RISE Research Institutes of Sweden, Smart Hardware, Bio- and Organic Electronics, Södra Grytsgatan 4, 602 33 Norrköping, Sweden; mats.sandberg@ri.se

**Keywords:** ZnO nanorods, aqueous chemical growth, screen printing, PEDOT:PSS cathode, carbon fibers, photocatalytic fuel cells (PCFCs), salicylic acid, pollutant removal

## Abstract

Water pollutants harm ecosystems and degrade water quality. At the same time, many pollutants carry potentially valuable chemical energy, measured by chemical oxygen demand (COD). This study highlights the potential for energy harvesting during remediation using photocatalytic fuel cells (PCFCs), stressing the importance of economically viable and sustainable materials. To achieve this, this research explores alternatives to platinum cathodes in photocathodes and aims to develop durable, cost-effective photoanode materials. Here, zinc oxide nanorods of high density are fabricated on carbon fiber surfaces using a low-temperature aqueous chemical growth method that is simple, cost-efficient, and readily scalable. Alternatives to the Pt cathodes frequently used in PCFC research are explored in comparison with screen-printed PEDOT:PSS cathodes. The fabricated ZnO/carbon anode (1.5 × 2 cm^2^) is used to remove the model pollutant used here and salicylic acid from water (30 mL, 70 μM) is placed under simulated sunlight (0.225 Sun). It was observed that salicylic acid was degraded by 23 ±0.46% at open voltage (OV) and 43.2 ± 0.86% at 1 V with Pt as the counter electrode, degradation was 18.5 ± 0.37% at open voltage (OV) and 44.1 ± 0.88% at 1 V, while PEDOT:PSS was used as the counter electrode over 120 min. This shows that the PEDOT:PSS exhibits an excellent performance with the full potential to provide low-environmental-impact electrodes for PCFCs.

## 1. Introduction

Water is a basic requirement for life. At the same time, an increasing number of artificial substances are entering natural waters, threatening both aquatic life and the quality of sources for drinking water. The remediation of aquatic pollutants is complicated by their typically low concentrations, making their effective removal difficult and costly. A case in point is the difficulties encountered in the remediation of pharmaceutical substances, where even low concentrations can harm aquatic life. Consequently, there is a need for remediation techniques that are techno-economically feasible for low-concentration pollutants. In this light, photocatalysis is an interesting option [1,2]. The motivation behind employing photocatalytic fuel cells for wastewater treatment stems from a dual objective: not only the effective removal of pollutants but also the harvesting of their energy content. This innovative approach combines the principles of the photocatalytic degradation of pollutants and energy harvesting to address environmental concerns, while simultaneously tapping into the potential energy reservoir within pollutants. Combining wastewater treatment with an energy-generating process has the potential to create sustainable and economically viable solutions. This integration of photocatalytic fuel cells not only improves the efficiency of pollutant removal, but also contributes to the broader goal of resource optimization, offering a promising avenue for environmentally friendly wastewater treatment with added energy benefits.

The rapid advancement of materials and technologies for water purification has been ongoing for a long time. One of its main goals is to find a treatment that is efficient, low-cost, up-scales easily, and is environmentally friendly; the most challenging objective is its accessibility on a global scale.

The chemical energy carried by water pollutants can be harvested through oxidation processes. This is defined as the chemical oxygen demand (COD) of polluted water. The energy content of water pollutants can potentially be captured and used. Heidrich estimated the energy content of industrial and household wastewater to be 1 × 10^7^ J/m^3^ (137 × 10^17^ J/year). For the United States, that would translate into 4.4 × 10^17^ J, which corresponds to 2.8% of the total electricity consumption in the United States [3].

Salicylic acid is one of the major pollutants that is mainly found in the wastewater from our cosmetics, pharmaceutical, and food industries. Further, it is suitable as a model compound to test the remediation of pharmaceutical pollutants. One main, typical challenge with pharmaceutical pollutants is their very low concentration. This makes removal and destruction very expensive with conventional methods. Photocatalysis, on the other hand, is one of very few pollution-removal methods that can be viable for low-concentration pollutants [4,5].

Heterogeneous photocatalysis based on nanostructures has the full capability to meet this demand, and it has the potential to provide a comparatively simple and cost-effective solution. Currently, the photocatalytic purification of water under sunlight is an outstanding approach in comparison to other traditional techniques [6,7]. Typically, there are three main phases of any photocatalytic reaction: (i) Light (photons) strike on the surface of the semiconductor, and if the incident photons have greater energy than the semiconductor’s bandgap energy, then the valance electron will obtain enough energy to jump into the conduction band. (ii) Holes (electron vacancies) in the valance band of the semiconductor will oxidize donor molecules and produce hydroxyl radicals when they react with water molecules. These hydroxyl radicals have a very strong oxidization power to degrade pollutants. (iii) In the conduction band, the electrons will react with dissolved oxygen groups to produce superoxide ions. In this fashion, the electrons (in the conduction band) and holes (in the valence band) will undergo ongoing oxidation and reduction reactions with various species on the semiconductor’s surface [8].

Photocatalytic fuel cells (PCFCs) can remediate pollutants from wastewater while harvesting the chemical bond energy of the pollutants [9]. The motivation behind employing photocatalytic fuel cells for wastewater treatment stems from a dual objective: the effective removal of pollutants and the harnessing of their energy content. This innovative approach combines the principles of photocatalysis and energy harvesting to address environmental concerns, while simultaneously tapping into the potential energy reservoir within pollutants. By generating energy from a wastewater treatment process, a sustainable and economically viable solution can be provided. This integration of photocatalytic fuel cells not only improves the efficiency of pollutant removal but also contributes to the broader goal of resource optimization, offering a promising avenue for environmentally friendly wastewater treatment with added energy benefits. Different PCFC configurations are reported for different applications, for example, double-photoelectrode PCFCs, single-photoelectrode PCFCs, dual rotating-disk PCFCs, optofluidics based micro-PCFCs, and air-cathode PCFCs are reported for the treatment of wastewater, generation of electricity, and hydrogen production [10].

The PCFCs have several advantages over conventional microbial fuel cells. For example: (i) PCFC systems utilize sunlight as an energy source which leads to the immediate generation and transfer of electrons and holes and, as a result, it boosts the degradation efficiency of pollutants; (ii) it is possible to design/develop advanced and efficient photocatalytic nanomaterials; (iii) PCFCs can be fabricated through simple, cheap, and scalable approaches and can be used easily in limited-reaction circumstances; and (iv) PCFC systems have excellent oxidation ability through photogenerated holes and OH ions [11]. PCFC systems have been previously used for the degradation of a broad range of pollutants, for example, dyes, antibiotics, heavy metal ions, and alcoholic compounds [12].

Advancements in composite light harvester materials and strategies for design optimization are needed for an efficient photocatalysis process. Materials with a large surface area are required because heterogeneous chemical reactions need a greater contact surface between the material and fluid. When sunlight is selected as a source of radiation to treat water, there is a need for photostable and nontoxic semiconducting materials that can harvest the maximum amount of sunlight.

Among the most investigated photocatalytic materials (CeO_2_, WO_3_, Fe_2_O_3_, GaN, CdS, and ZnS), semiconductor metal oxides (ZnO and TiO_2_) are cheap, efficient, abundant, and nontoxic materials. These can be fabricated through fast, simple, cost-effective, and scalable fabrication techniques like aqueous chemical growth (ACG), chemical vapor deposition (CVD), vapor–liquid–solid (VLS), and electron beam evaporation. Recently, the superior photocatalytic activities of TiO_2_ have been widely studied. It is a low-cost material swith good stability. But, it has limited applications due to low quantum yield with a high rate of charge carrier recombination [13,14,15,16,17,18,19].

ZnO is one of the most widely investigated photocatalysts [20]. This is an extremely photosensitive inert semiconductor, possessing a 3.37 eV bandgap energy and remarkable electron mobility, reaching 1000 cm^2^ V^−1^ s^−1^ for individual nanowires [21], which provides fast electron transport and decreases the recombination of charge carriers. ZnO exhibits good conductivity for both electrons and holes when compared to other oxide materials. The main advantage of ZnO over TiO_2_ is its ability to absorb a larger fraction of sunlight. Several reports have demonstrated that the modified ZnO nanowires (NWs) demonstrate superior photocatalytic properties along with other elements [22]. 

ZnO NRs provide several advantages because they offer a high surface area leading to a much higher surface area for photocatalytic reactions. It also enhances light scattering for improved absorption and more effective charge separation and collection due to the closeness between the photogenerated charge carriers and the interface (NWs-electrolyte). The photocatalytic properties of different ZnO nanostructures have been studied, for example, NWs, nanorods (NRs), nanodiscs, nanowalls, and microspheres [23,24]. It considerably decreases the charge-carrier diffusion length and enhances the surface evolution kinetics. The photocatalytic properties of ZnO can be enhanced by constructing composites with carbon-based materials (carbon fiber/ZnO NRs). Further, it is also estimated that by depositing the NWs on carbon fiber, the surface area can increase by at least 400,000 times compared to a thin film deposited on the fiber.

Carbon fiber/ZnO nanostructured composites have been previously investigated for various applications, including photocatalysis, environmental remediation, and energy harvesting [25,26,27]. Carbon fibers (CFs) have good flexibility and chemical resistivity with higher electroconductivity and great mechanical strength [28,29,30].

Carbon fibers have been extensively studied as a substrate material to develop water and gas treatment/purification systems [31,32,33]. Carbon fiber substrates are also used in the development of photocatalytic devices as a structured support material and have shown a much better photocatalytic performance as compared to other substrates [34,35,36]. In this way, the carbon fibers are not only providing their surface for the deposition of the photocatalytic materials, but their high conductivity and absorptivity are also contributing to enhancing the overall photocatalytic activity of the composite material [25,37,38]. For the reasons stated above, ZnO nanorods grown on carbon fibers as photo-anode materials are deemed techno-economically feasible for the PCFC concept.

In this study, densely packed ZnO nanorods on carbon fibers were fabricated using a straightforward and low-temperature aqueous chemical method. This strategy was used to produce functional carbon fibers in an easy, facile, and cost-effective manner, which is suitable for large-scale manufacturing. The photocatalytic degradation of salicylic acid in water under the simulated sunlight at 0 and 1 V applied voltage was demonstrated. The morphology, structure, elemental composition, and optical properties of the synthesized material were analyzed using scanning electron microscopy (SEM), energy-dispersive X-ray spectroscopy (EDS), and UV–visible spectroscopy.

Replacing expensive and scarce noble metal catalysts such as platinum is a prerequisite for the techno-economical feasibility and sustainability of PCFCs. For this reason, PEDOT:PSS (poly(3,4-ethylenedioxythiophene) polystyrene sulfonate) is investigated as an alternative low-cost and sustainable cathode catalyst for PCFCs. Moreover, PEDOT:PSS compositions can be deposited by scalable and low-cost methods. Here, PEDOT:PSS’s electrode characteristics were compared to those of the platinum electrodes in PCFCs. 

## 2. Materials and Methods

### 2.1. Materials

Commercially purchased carbon fibers (from ABIC Kemi, Norrköping, Sweden) with a diameter of 8 ± 1 μm were acquired in the form of fabric sheets. The fabric sheet samples were cut to a size of 2 × 2 cm^2^. Coatings on the fiber surface were removed by exposing the fibers to a flame followed by ultrasonication in ethanol and DI water for 20–50 min. 

Zinc nitrate hexahydrate (Zn(NO_3_)_2_ 6H_2_O) 99%, zinc acetate dihydrate (Zn(CH_3_COO)_2_ 2H_2_O) 99.0%, hexamethylenetetramine (HTMA, C_6_H_12_N_4_) 99%, and salicylic acid (2(HO)C_6_H_4_CO_2_H) were purchased from Sigma-Aldrich. All chemicals used in this study were of analytical grade with a purity of 99% and no further purification was necessary. The PEDOT:PSS ink (from Heraeus with conductivity ∼1000 S/cm) was screen printed on polyethylene terephthalate (PET) substrate at Printed Electronics Arena (PEA), RISE Research Institutes of Sweden, Norrköping, Sweden, with a film thickness of ∼1.0 μm.

### 2.2. Growth of ZnO NWs on Carbon Fibers

Densely packed ZnO nanorods are produced on carbon fiber surfaces employing an economical, low-temperature (<100 °C) aqueous chemical growth method, as outlined by Vayssieres et al. [39]. The process involves depositing a ZnO seeding layer on the carbon fiber surface prior to nanorod fabrication, following the method introduced by Womelsdorf et al. [40]. In the initial step, a solution containing 0.01–0.7 M zinc acetate dihydrate in methanol was heated to 55–85 °C for 1–3 h with continuous magnetic stirring. Subsequently, 0.02–0.06 M KOH was slowly added. Carbon fibers were immersed in this solution for 2–5 min to achieve a thin and uniform ZnO seeding layer, followed by air drying. The growth solution comprised equimolar amounts (0.15–0.70 M) of zinc nitrate hexahydrate (Zn(NO_3_)_2_·6H_2_O) and hexamethylene tetramine (HMT and C_6_H_12_N_4_) in DI water was then prepared. Lastly, the ZnO-seeded carbon fibers were placed in an electric furnace at 60–95 °C for 2–5 h to facilitate the growth of ZnO nanorods through specific reactions [41]. 

In the 1st step, the ammonia was produced due to the reaction between HMT (C_6_H_12_N_4_) and water, where
(CH_2_)_6_ N_4_ + 6H_2_O → 6HCHO + 4NH_3_(1)

In the 2nd step, the reaction between ammonia and water produces ammonium and hydroxide ions, where
NH_3_ + H_2_O → NH_4_^+^ + OH^−^(2)

In the 3rd step, the reaction between hydroxide ions with zinc ions produce solid ZnO NRs on the surface of carbon fibers, where
2OH^−^ + Zn^+2^ → ZnO_(s)_ + H_2_O(3)

The photocatalytic reactions were performed using an H-shaped PEC cell under the simulated sunlight of 0.225 Sun. The simulated sunlight was directed on the surface of the working electrode (carbon fiber/ZnO NRs fabric sheet with a size of 1.5 × 2 cm^2^) as shown in schematic Figure 1. The H-shaped PEC cell consisted of two parts which were separated by an ion-exchange membrane. Salicylic acid solution (30 mL, pH 6.0) with a concentration of 70 μM was filled in both chambers of the H-shaped fuel cell. The photodegradation was monitored every 40 min for 2 h at 0 and 1 V and samples with a volume of 1 mL were collected after every 20 min for UV–visible absorption spectroscopy measurements. Platinum electrodes (surface area 9.4473 cm^2^) and PEDOT: PSS-coated film (surface area 6.0 cm^2^) electrodes were used as counter electrodes for comparative studies. The voltage was applied through a Keithley 2400 source meter, and the photocurrent was measured. 

### 2.3. Measurements

The morphology, structure, and elemental composition of the fabricated ZnO nanorods were investigated through scanning electron microscopy (SEM, Sigma 500 Gemini equipped with EDS). The SEM images were obtained using the In-lens detector, which primarily gathered secondary electrons, offering topographical details of the sample at magnifications ranging from 250× to 1,000,000. The working distance stood at 5.0 mm, operated under a voltage (ETH) of 1.0 kV.

Elemental analysis was performed by utilizing energy-dispersive X-ray spectroscopy (EDS) in conjunction with the SEM.

Absorption spectroscopy investigations were conducted at room temperature using the Absorption Spectrometer Lambda 900 (232003—Lambda 900 PerkinElmer). The salicylic acid showed a maximum absorption peak at ~285 nm. This peak was used to investigate the change in the absorbance peak of salicylic acid at a maximum wavelength of 285 nm and DI water was used as a reference.

## 3. Results and Discussions

The top section of Figure 1 illustrates the primary steps involved in the decoration of carbon fibers with ZnO nanorods. The functionalization of carbon fibers with ZnO nanorods was achieved through a straightforward and low-temperature aqueous chemical growth process. Figure 1 (bottom) shows the schematic illustration of an H-shaped PEC cell (divided into two sections by glass frit with a carbon anode and a PEDOT:PSS counter electrode. A glass frit was used between both sections as an ion-transfer membrane. The current ran through an external circuit connected to a Keithley 2400 source meter.

Uniform and densely arranged ZnO nanorods on carbon fibers, fully covered by vertically aligned ZnO nanorods with hexagonal structures, were prepared. The ZnO nanorods exhibit a uniaxial orientation of 0001 concerning the carbon fiber surface. SEM images in Figure 2 reveal the apparent average diameter and length of the fabricated ZnO nanorods, estimated at approximately 100–250 nm and 800–1200 nm, respectively. SEM images give insights into the size distribution and morphology of the fabricated nanorods. The precise control over the size, orientation, and arrangement of the nanorods is crucial for tailoring the material’s properties for various applications, such as in electronics, sensors, catalysts, or energy storage.

Figure 3 displays the EDS spectra and elemental mapping images of the ZnO NRs/carbon fibers with the analysis of chemical compositions by percentage. It was applied to determine the chemical composition of grown ZnO NRs. The EDS analysis confirms the presence of carbon (C), oxygen (O), and zinc (Zn) elements, affirming the absence of any other impurities in the ZnO NRs/carbon fiber sample. The EDS spectral peaks of Zn are located at 1.0 and 0.88 keV, and for O it is centered at 0.5 keV, and for C it is positioned at 0.3 keV. These peaks are distinctive and arise from the interaction of X-rays with the atoms in the sample. The strong intensity of the Zn and O peaks is attributed to the dense coverage of ZnO NRs on the carbon fibers. Figure 3 displays elemental mapping, indicating the presence of C, O, and Zn atoms in the ZnO NRs/carbon fiber sample. It verifies the uniform distribution of O and Zn elements across the entire carbon fiber. This uniformity is crucial for ensuring consistent material properties and performance. Atomic and weight percentages of C, O, and Zn elements are tabulated in Figure 3, providing quantitative data on the elemental composition of the sample. These percentages offer valuable information for understanding the stoichiometry and overall makeup of the ZnO NRs/carbon fiber composite. Overall, the EDS results show a detailed characterization of the chemical composition and spatial distribution of elements within the ZnO NRs/carbon fibers, confirming their successful fabrication and providing essential insights for further analysis and optimization.

Figure 4 shows a schematic illustration of charge generation, transfer, and pollutant degradation at the ZnO/carbon-fiber anode and the PEDOT:PSS cathode under sunlight. The proposed mechanism for the fabricated fuel cell is as follows:(4)$$\mathrm{ZnO}+\mathrm{Sunlight}\to \mathrm{ZnO}\;\left(e_{\mathrm{CB}}^{-}+h_{\mathrm{VB}}^{+}\right)$$
(5)$$H_{2}O+{\;h}^{+}\to {\mathrm{OH}}^{●}\left(\mathrm{hydroxyl}\;\mathrm{radical}\right)+H^{+}$$
(6)$$\mathrm{Salicylic}\;\mathrm{acid}+{\;\mathrm{OH}}^{●}\to {\mathrm{Oxidation}\;\mathrm{Product}\;\mathrm{CO}}_{2}+H_{2}O$$
(7)$$O_{2}+{\;e}^{-}\to {\;O}_{2}^{-}\left(\mathrm{superoxide}\;\mathrm{anion}\right)$$
(8)$$O_{2}^{-}+{\;H}_{2}O\;\to {\;H}_{2}O_{2}+{\;\mathrm{OH}}^{●}$$
(9)$$\mathrm{Salicylic}\;\mathrm{acid}+{\;\mathrm{OH}}^{●}\to \left(\mathrm{Oxidation}\;\mathrm{of}\;\mathrm{Salicylic}\;\mathrm{acid}\right){\;\mathrm{CO}}_{2}+H_{2}O$$

As sunlight shines on the surface of the ZnO NRs, the absorbed photons provide enough energy for electrons in the valence band to transition to the conduction band, creating holes in the valance band. The interaction between holes and water at the surface of the ZnO NRs yields hydroxyl radicals (^●^OH), which act as strong oxidizing agents to reduce the pollutant (salicylic acid) present in the water. Electrons are transferred toward the counter electrode through the carbon fibers, where they react with oxygen and subsequently reduce it into superoxide anions (O^2−^), where either these anions directly attack organic pollutants in water, or they react with water to produce hydroxyl radicals (^●^OH), which again reduce the pollutant in the water.

Figure 5 shows the UV–visible spectral time scans of salicylic acid solutions collected after every 40 min (0–120 min) illuminated under the simulated sunlight (0.25 Sun). It shows a continuous degradation of salicylic acid with time. There is an exponential decrease in the concentration of salicylic acid with the illuminated time (0–120 min). The application of 0 V and 1 V potential across the electrodes, while using platinum and PEDOT:PSS film as counter electrodes, reveals that the degradation rate of salicylic acid is approximately doubled when a potential of 1 V between the electrodes as compared to 0 V is applied. Figure 5e shows UV–visible spectral time scans of the control sample (carbon fibers without ZnO NWs) at 1 V applied potential with a Pt counter electrode and there is no degradation of salicylic acid after 120 min (about 2 h). 

Figure 5f shows a plot of time verses percentage degradation for salicylic acid. It shows that the PEDOT:PSS counter electrode has a very good performance, and it competes with the Pt electrode very well. Therefore, screen-printed PEDOT:PSS film exhibits the full potential to contend with Pt electrodes and can compete with Pt electrodes while providing low-environmental-impact electrodes for PCFCs.

PEDOT:PSS cathodes show promise across various fields due to the conducting polymer’s high electrical conductivity, flexibility, and environmental stability, presenting an alternative to metals like platinum. These cathodes efficiently catalyze electrochemical reactions such as oxygen reduction reactions (ORRs) in fuel cells and metal–air batteries, with comparable activity to platinum, especially in alkaline conditions, making them suitable for renewable energy applications. Additionally, PEDOT:PSS exhibits durability and corrosion resistance, ensuring long-term stability in electrochemical systems. Beyond energy conversion, PEDOT:PSS cathodes have applications in sensors, actuators, and electronic devices. Their compatibility with flexible substrates and solution-based processing enables the fabrication of lightweight, flexible electrochemical devices for wearable electronics and biomedical applications. Moreover, their biocompatibility and ease of functionalization make them suitable for interfacing with biological systems, paving the way for bioelectronic applications and biosensors. However, further research is necessary to optimize PEDOT:PSS’s catalytic activity, stability, and scalability compared to traditional metal catalysts like platinum. Comparative studies with other non-precious metal catalysts and advanced materials are also essential to understand PEDOT:PSS’s advantages and limitations in various electrochemical applications. Integrating PEDOT:PSS cathodes offers exciting opportunities for advancing electrochemical technologies and addressing sustainability and cost-effectiveness challenges.

The degradation (%) of salicylic acid was calculated at 283 nm using the absorbance values according to the following relation:Degradation (%) = 100 × (1 − A_t_/A_0_)(10)
where, A_t_ is the absorbance after time t minutes, and A_0_ is the absorbance at zero time. It is found that the salicylic acid is degraded 23 ± 0.46% at OV and 43.2 ± 0.86% at 1 V in 120 min when the Pt mesh was used as a counter electrode while it was degraded 18.5 ± 0.37% at OV and 44.1 ± 0.88% at 1 V in 120 min when PEDOT:PSS was used as a counter electrode.

The SEM images of the fabricated ZnO nanowires (NWs) on carbon fibers post-photocatalytic measurements are depicted in Figure 6. The SEM images reveal that the NWs show no signs of degradation or damage. 

Table 1a,b shows the measured photocurrent and dark current for 2 h at 0 V and 1 V applied potential with the Pt counter electrode, and Table 1c,d shows the measured photocurrent and dark current at 0 V and 1 V applied potential with the PEDOT:PSS counter electrode. It was found that the PEDOT:PSS electrode shows very good electrode characteristics compared to platinum electrodes and exhibits full potential to provide efficient electrodes for PCFCs.

## 4. Conclusions

In summary, this research work shows the fabrication of highly dense ZnO NRs on the surface of carbon fibers with a low-cost, simple, and scalable aqueous chemical process. The fabricated ZnO NRs/carbon fibers are highly photocatalytic under simulated sunlight. The electrode’s durability allows for practical use in a variety of polluted water matrices. The ZnO NW carbon electrodes may be applied in other devices, for example, piezoelectric energy harvesters, UV/strain/chemical/biosensors, solar cells, light-emitting diodes, and charge storage devices. Importantly, the finding that PEDOT:PSS functions as a low-cost alternative to noble metal catalysts in PCFC cathodes takes the concept a step further towards its possible implementations. This research work not only presents compelling results regarding the fabrication and performance but also contributes valuable insights to the existing literature in the field. Moving forward, further exploration of the applications and optimization of these materials could lead to even more impactful advancements in environmental remediation and beyond.

## Figures and Tables

**Figure 1 nanomaterials-14-00636-f001:**
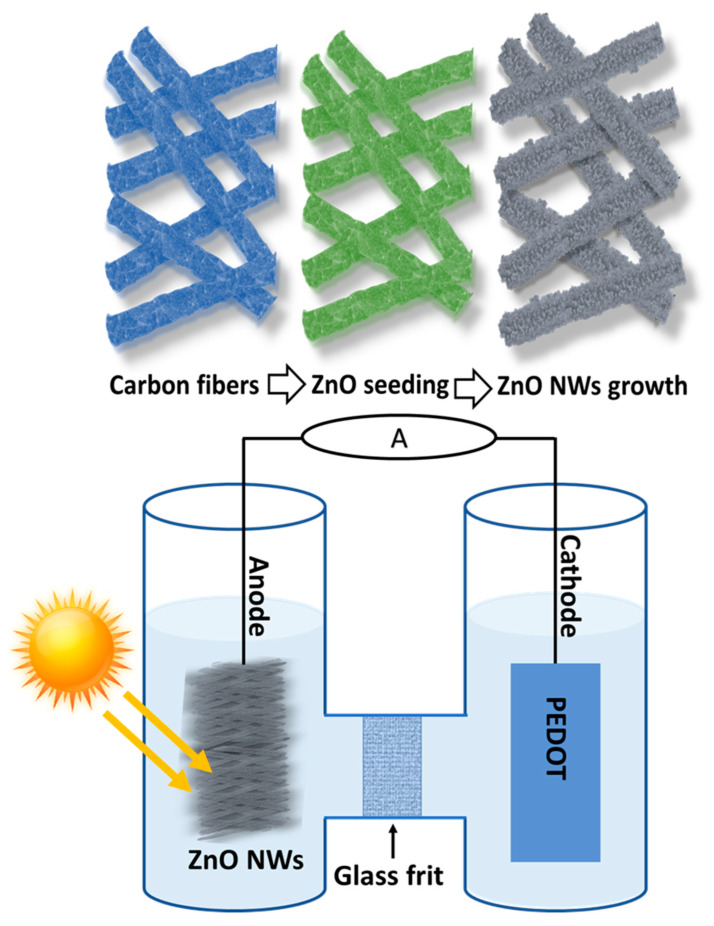
Schematic illustration of an H-shaped photoelectrochemical cell with a carbon fiber/ZnO NW anode and a PEDOT:PSS cathode, separated into two parts through a glass frit. The current flows through an external circuit (Keithley source meter 2000).

**Figure 2 nanomaterials-14-00636-f002:**
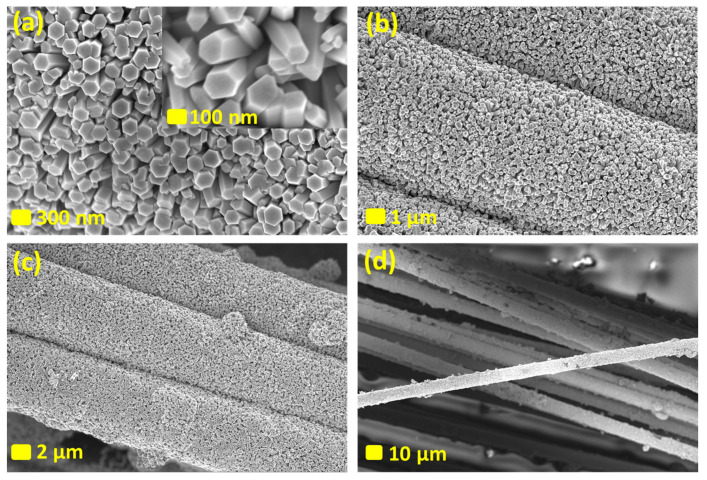
(**a**–**d**) SEM images of ZnO NRs fabricated on the surface of carbon fibers at different scales. The scale bars are specified on every figure.

**Figure 3 nanomaterials-14-00636-f003:**
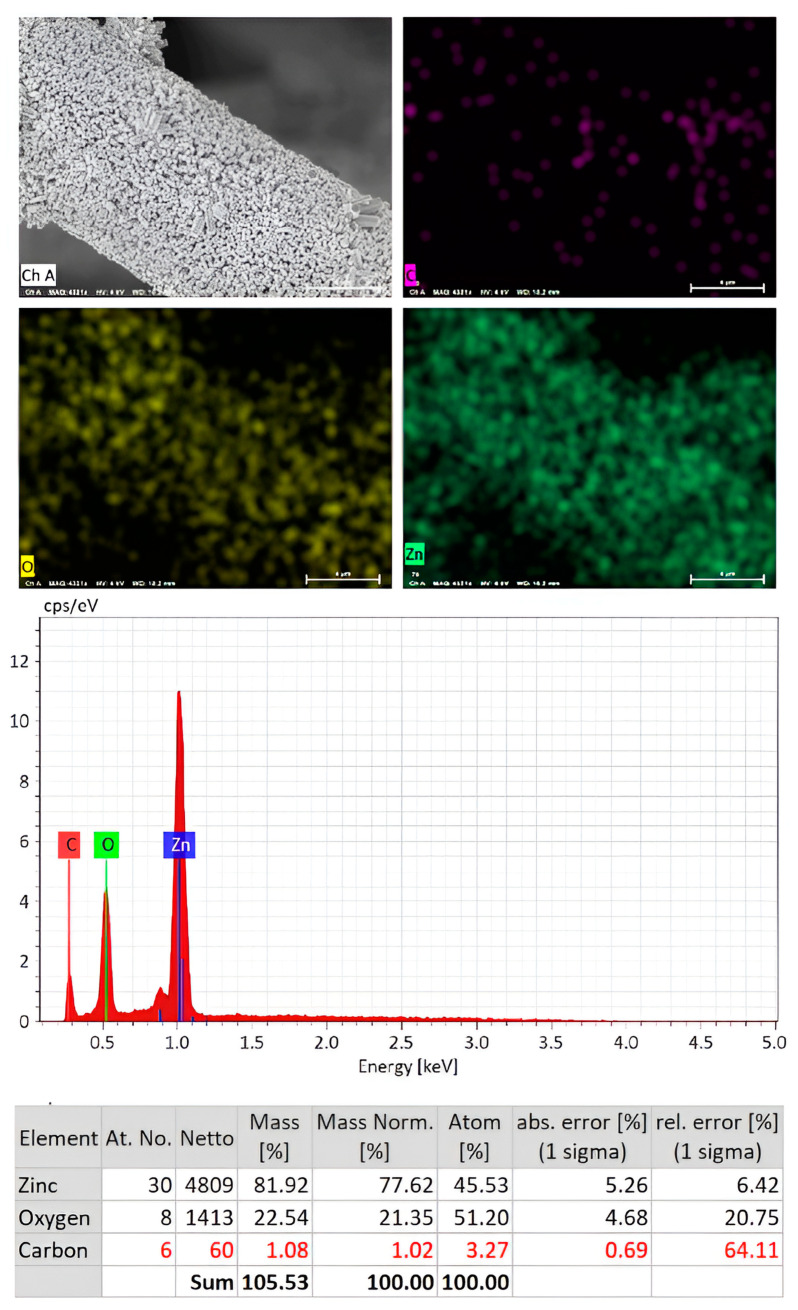
Elemental mapping and EDS spectra of the fabricated ZnO NRs on carbon fiber displays elemental mapping images of oxygen and zinc atoms. The atomic and weight percentages of Zn, O, and C elements are also tabulated.

**Figure 4 nanomaterials-14-00636-f004:**
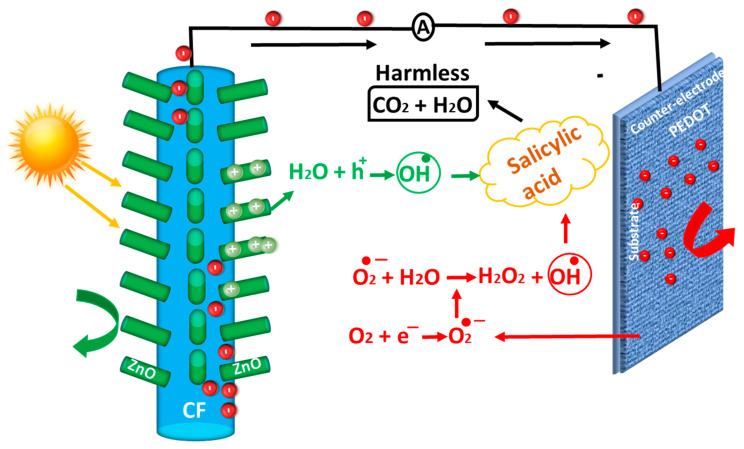
Schematic illustration of charge generation, transfer, and pollutant degradation at ZnO/carbon-fiber anode and PEDOT:PSS cathode under the sunlight.

**Figure 5 nanomaterials-14-00636-f005:**
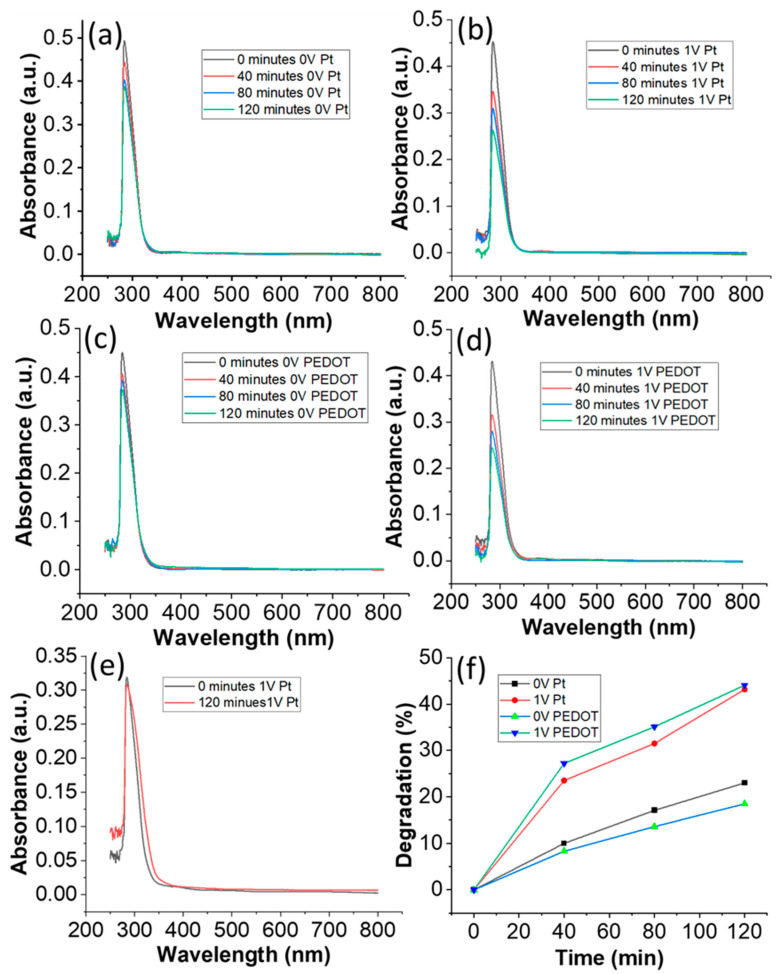
UV–visible spectral time scan of salicylic acid solutions collected after every 40 min (0–120 min, (**a**) 0 V applied potential while using platinum as a counter electrode, (**b**) 1 V applied potential with Pt counter electrode, (**c**) 0 V applied potential while using PEDOT:PSS as a counter electrode, (**d**) 1 V applied potential while using PEDOT:PSS as a counter electrode, (**e**) control sample (carbon-fibers without ZnO NWs), 1 V applied potential with a Pt counter electrode, and (**f**) graph of time verses percentage degradation for salicylic acid.

**Figure 6 nanomaterials-14-00636-f006:**
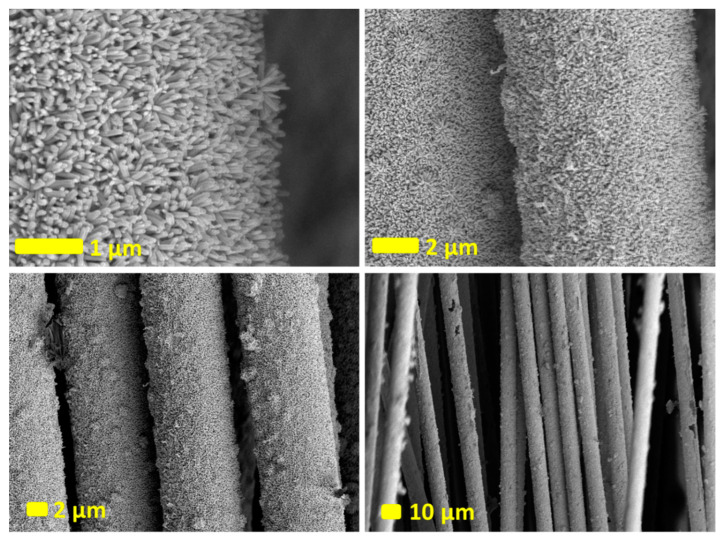
SEM images of fabricated ZnO NRs on the surface of carbon fibers after the photocatalytic measurements.

**Table 1 nanomaterials-14-00636-t001:** (**a**,**b**) shows measured photocurrent and dark current at 0 V and 1 V applied potential with Pt counter electrode, and (**c**,**d**) shows measured photocurrent and dark current at 0 V and 1 V applied potential with PEDOT:PSS counter electrode.

**(a)** **0V**			**(b)** **1V**		
**Time (min)**	**Current** **(μA)**	**Dark current (μA)**	**Time (min)**	**Current** **(μA)**	**Dark current** **(μ****A)**
0	21.6 ± 0.1	4.0 ± 0.1	0	74.09 ± 0.1	32.01 ± 0.1
40	19.1 ± 0.1	3.8 ± 0.1	40	64.0 ± 0.1	25.6 ± 0.1
80	18.53 ± 0.1	3.7 ± 0.1	80	57.03 ± 0.1	20.3 ± 0.1
120	17.31 ± 0.1	3.6 ± 0.1	120	53.01 ± 0.1	16.21 ± 0.1
**(c)** **0V**			**(d)** **1V**		
**Time (min)**	**Current** **(μ****A)**	**Dark current** **(μ****A)**	**Time (min)**	**Current** **(μ****A)**	**Dark current** **(μ****A)**
0	9.1 ± 0.1	1.6 ± 0.1	0	39.56 ± 0.1	17.6 ± 0.1
40	8.3 ± 0.1	1.5 ± 0.1	40	35.93 ± 0.1	14.5 ± 0.1
80	8.02 ± 0.1	1.5 ± 0.1	80	32.59 ± 0.1	12.7 ± 0.1
120	7.95 ± 0.1	1.4 ± 0.1	120	31.05 ± 0.1	12.1 ± 0.1

## Data Availability

The data that support the findings of this study are available upon reasonable request from the authors.

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
