# Peer review of "Sustainable and Low-Cost Electrodes for Photocatalytic Fuel Cells"

_nanomaterials, 2024, doi:10.3390/nano14070636_

Round 1

Reviewer 1 Report

Comments and Suggestions for Authors

This work regards high dense zinc oxide nanorods deposited on carbon fiber surfaces using a low-temperature aqueous chemical growth method. Alternatives to the typically used Pt are explored in comparison with screen-printed PEDOT:PSS cathodes. The ZnO/Carbon anode realized in this work is used to remove the salicylic acid from water under simulated sunlight. Such samples presented a photocatalytic activity. 

The manuscript needs to be revised and the following main points need to be carefully addressed before it can be considered for publication: 

1.     First, the added value of this review paper among the large literature in the field should be clearly addressed in the manuscript. 

1.     The introduction needs to be improved. Before mentioning the advantages of some synthesis methods (line 108), it is necessary to mention several synthesis technologies used in the literature for the fabrication of ZnO nanorods, including at least the carbo-thermal transport growth [DOI:10.1007/s00339-007-3946-4] and the electron beam evaporation [DOI10.1016/j.mssp.2017.08.015]. 

2.     In the paragraph “2. Materials and methods” some information and measurement conditions are missing. About the SEM observations, the brand (Zeiss) of the SEM, and the parameters used in the measurements (such as the acceleration voltage, the working distance used and so on) should be reported. Also, the EDS system brand and model are missing. 

3.     Fig.2: The four images must be clearly identified by the labels (a), (b), (c) and (d) on the figure. Furthermore, in the text (line 235) there is a range for the diameter values and one for the length of the nanorods (NRs), however there is no evidence from the images about the length of the NRs. Therefore, an additional image (cross section or otherwise) should be included in the manuscript to clearly show the length of the NRs. 

4.     Fig.3: the EDS table of the elements present in the sample shows a sum greater than 100% (105.53%) which has no meaning. Authors should carefully correct and comment on the text. 

5.     Regarding the degradation calculation reported in Fig. 5(f), the errors should be included both in the calculation and consequently as error bars in the figure. 

6.     Fig. 6: this figure should allow a comparison with the two situations before and after the photocatalytic exposure, and this can be done by comparing Fig.2 with fig 6. It is therefore necessary to show in Fig. 6 the images with the same magnification as those in Fig. 2. For example, Fig. 2(a) appears to be the image with the highest magnification (300 nm marker), in Fig. 6 there should be a corresponding image with the same magnification. Similarly, for (b), (c) and (d). Furthermore, it is mandatory to show images at higher magnifications to clearly analyze the real effect of photocatalytic activity on the samples and to support the conclusions about this comparison. 

7.     Table 1: the errors in the values reported should be included. 

8.     Moreover, the conclusions should highlight first the main results of the present work and then the value that the manuscript adds to the current literature in the field along with some perspectives. 

9.     All references in the list need to be completed with the DOI. 

Comments on the Quality of English Language

The manuscript should be revised by an English native speaker.

Author Response

We want to thank the reviewer for his valuable comments. We have made the possible revisions and clarifications in the revised manuscript. We are pleased to inform you that a native English speaker has thoroughly reviewed the manuscript, significantly improving its language and readability. Changes in the manuscript are highlighted in yellow. The PDF file uploaded serves as a comprehensive reply to all comments and requests.

Reviewer 2 Report

Comments and Suggestions for Authors

In the manuscript entitled as "Sustainable and Low-Cost Electrodes for Photocatalytic Fuel 2Cells" investigated a sustainable and low-cost electrode material for photocatalytic fuel cells. The researchers prepared electrodes using zinc oxide (ZnO) nanorods on carbon fibres and oxidised pollutants in water into energy via a photocatalytic reaction. The results show that this material degrades well under simulated sunlight and that the degradation rate is about twice as high when 1 V is applied as when 0 V is applied.

This article represents a valuable study which provides the growth of ZnO nanorods on carbon fibres and their application to photocatalytic fuel cells (PCFC). When sunlight strikes the surface of ZnO nanorods, the absorbed photon energy is sufficient to cause electrons in the valence band to jump into the conduction band, creating holes in the equilibrium band. The interaction of the holes with water on the surface of the ZnO nanorods produces hydroxyl radicals (-OH), which are strong oxidising agents that degrade pollutants in water (salicylic acid). Electrons are transported through the carbon fibres towards the counter electrode, where they react with oxygen and reduce it to superoxide anions (O2-), which either attack the organic pollutants in the water directly or react with the water to produce hydroxyl radicals (-OH), which again degrade the pollutants in the water.In any case, despite its qualities, the paper has shortcomings which must be addressed before the manuscript can be published.

1.         The structure of the article could be further optimised to present the experimental process and results more clearly. For example, the electrode preparation and measurement methods could be described in detail in the methods section, as well as the morphology, structure and optical properties of ZnO nanorods could be analysed more systematically in the results section.

2.         In the results section, a more in-depth analysis of the experimental data can be carried out to better understand the growth process of ZnO nanorods on carbon fibres and the factors affecting their photocatalytic performance.

3.         In the discussion section, the results of this study can be compared with other related studies to better demonstrate the innovativeness and strengths of this study. For example, PEDOT:PSS is mentioned in the article as an alternative to PCFC cathode catalysts, but the discussion of its performance and application prospects is rather brief, and there is a lack of comparisons and evaluations of it with other catalysts.

4.         The concluding part of the article is rather brief and lacks a comprehensive summary of the experimental results and an outlook on future research directions.

In conclusion, the paper is recommended to be published after minor revision.

Author Response

(The authors gave the same response as above.)

Round 2

Reviewer 1 Report

Comments and Suggestions for Authors

The authors have addressed the required issues improving the manuscript. Therefore, it is now suitable for publication.